# Breathing Practices for Stress and Anxiety Reduction: Conceptual Framework of Implementation Guidelines Based on a Systematic Review of the Published Literature

**DOI:** 10.3390/brainsci13121612

**Published:** 2023-11-21

**Authors:** Tanya G. K. Bentley, Gina D’Andrea-Penna, Marina Rakic, Nick Arce, Michelle LaFaille, Rachel Berman, Katie Cooley, Preston Sprimont

**Affiliations:** 1Health and Human Performance Foundation, Los Angeles, CA 90272, USArachelberman1@gmail.com (R.B.); cooley.katherine@gmail.com (K.C.);; 2Neurosciences Graduate Program, University of California San Diego, La Jolla, CA 92093, USA; 3Department of Kinesiology, California State University Fullerton, Fullerton, CA 92831, USA

**Keywords:** breathing, breathwork, pranayama, respiration, diaphragmatic, stress, anxiety

## Abstract

Anxiety and stress plague populations worldwide. Voluntary regulated breathing practices offer a tool to address this epidemic. We examined peer-reviewed published literature to understand effective approaches to and implementation of these practices. PubMed and ScienceDirect were searched to identify clinical trials evaluating isolated breathing-based interventions with psychometric stress/anxiety outcomes. Two independent reviewers conducted all screening and data extraction. Of 2904 unique articles, 731 abstracts, and 181 full texts screened, 58 met the inclusion criteria. Fifty-four of the studies’ 72 interventions were effective. Components of effective and ineffective interventions were evaluated to develop a conceptual framework of factors associated with stress/anxiety reduction effectiveness. Effective breath practices avoided fast-only breath paces and sessions <5 min, while including human-guided training, multiple sessions, and long-term practice. Population, other breath paces, session duration ≥5 min, and group versus individual or at-home practices were not associated with effectiveness. Analysis of interventions that did not fit this framework revealed that extensive standing, interruptions, involuntary diaphragmatic obstruction, and inadequate training for highly technical practices may render otherwise promising interventions ineffective. Following this evidence-based framework can help maximize the stress/anxiety reduction benefits of breathing practices. Future research is warranted to further refine this easily accessible intervention for stress/anxiety relief.

## 1. Introduction

The rising prevalence of anxiety and chronic stress plagues populations worldwide. Prior to the COVID-19 pandemic, anxiety disorders had already attained startlingly high proportions, affecting nearly one-third of Americans at some point during their lifetime [1]. Since then, anxiety disorders have surged by an estimated 25.6% globally [2] and they burden the economy with healthcare costs and reduced productivity [3,4]. Both above and below the clinical threshold, anxiety and chronic stress impact physical, mental, and cognitive health, increasing the risk of cardiometabolic disorders, cancer, mental illness, neurodegenerative disease, and all-cause mortality [5,6,7,8,9,10,11,12,13,14]. Chronic activation of the stress response induces allostatic overload, a cumulative “wear and tear” on the brain and body that erodes resiliency and health [15]. 

Common stress and anxiety treatments rely on external factors, such as a therapist or medication, and come with modest effect sizes [16] or unfavorable side effects [17]. Effective, accessible, and risk-free solutions are needed. 

Voluntary regulated breathing practices may offer such a solution. These practices draw on both modern scientific studies and ancient yogic-pranayama concepts, with applications ranging from clinical anxiety treatment in adults to reducing stress reactivity in college athletes [18,19,20,21,22]. Common voluntary regulated breathing practices include diaphragmatic breathing, paced slow breathing, breathing with biofeedback, and alternate-nostril breathing (ANB). Breathing practices, when used in isolation, have the advantage of being universally accessible, scalable, and cost-free. They are not limited by access to healthcare services nor burdened by side effects and put potential treatment tools in the hands of the individual [23,24,25]. 

Breathing practices’ effects on the autonomic nervous system and brain may underlie their stress-reducing benefits. Effective breathing interventions support greater parasympathetic tone, which can counterbalance the high sympathetic activity intrinsic to stress and anxiety. Respiratory entrainment of brain rhythms offers an additional avenue through which breathing may influence neural circuit dynamics, cognition, and mood [26,27]. Breathing also uniquely engages in a reciprocal relationship with stress and anxiety, whereby stress and anxiety can both affect [28,29,30,31,32,33] and be affected by [20,21,22,34] altered respiratory patterns. Due to the shared neurophysiological underpinnings of stress and anxiety, from hereon, the term “stress” is used to encompass both conditions.

Although existing literature confirms the widespread benefits of slow breathing practices for stress/anxiety reduction [23,24,25,35,36,37,38], null or negative results are reported in some studies [39,40,41,42,43,44,45,46,47,48,49,50]. Discerning the distinguishing features between effective and ineffective interventions is of interest. 

We conducted a systematic review of clinical studies that evaluated the effectiveness of voluntary regulated breathing practices on psychometric measures of anxiety and stress. We identified intervention characteristics associated with effectiveness and used these data to build a conceptual framework of components associated with the stress reduction effectiveness of breathing practices. 

## 2. Materials and Methods

The Preferred Reporting Items for Systematic Reviews and Meta-Analysis (PRISMA) guidelines were strictly followed throughout. A protocol was not developed or registered.

### 2.1. Overview

We searched peer-reviewed, published literature for original clinical trials evaluating the effectiveness of voluntary regulated breathing practices on psychometric stress and anxiety measures. Our search covered articles published through March 2023, written in English, and from any geographic location. We operationally defined “voluntary regulated breathing practice” as any voluntary, regulated manipulation or control of the breath. We chose a broad and inclusive definition in an effort to capture a variety of regulated breathing methods, including deep breathing, diaphragmatic breathing, yoga or pranayama breathing, slow breathing, paced or patterned breathing, ANB, and more.

### 2.2. Search Strategy 

Our search strategy incorporated PubMed and ScienceDirect electronic searches (Section A.1) and manual reference-mining of previously published reviews and key clinical trials [22,23,24,25,33,35,36,37,38,51,52,53,54,55,56,57,58,59,60,61].

### 2.3. Selection Criteria

#### 2.3.1. Publication and Study Type

We included peer-reviewed, published randomized controlled trials (RCT) and other non-randomized prospective cohort and clinical studies. Gray literature, case studies, retrospective observational data analyses, editorials, and reviews were excluded. 

#### 2.3.2. Intervention

We included studies that considered any voluntary, regulated breathing practice as its own intervention, whether part of the treatment or control. We excluded interventions that used breath practices in combination with other interventions, such as yoga, meditation, tai chi, etc. We included interventions that used metronomes or visual/audio cues for guiding breathing pace and excluded those in which a device provided real-time biofeedback, such as data on expelled carbon dioxide (CO_2_), respiratory rate, heart rate variability (HRV), etc. We excluded interventions that incorporated means outside of participants’ direct control to manipulate or regulate participants’ oxygen or CO_2_ intake.

#### 2.3.3. Outcomes

We included studies where primary or secondary outcomes incorporated psychometric stress or anxiety measures, defined as those that quantitatively measure stress or anxiety by use of inventory, questionnaire, observation, or self-report. Studies that evaluated fear outcomes or physiologic measures were included only if psychometric measures of stress/anxiety were also included.

#### 2.3.4. Populations

The review included participants with clinical stress or anxiety (e.g., anxiety disorders, post-traumatic stress disorder) and non-clinical stress (e.g., healthy adults, youth, disease-specific populations). Studies with participants aged ≥1 year (i.e., capable of intentional breath regulation), of both genders and of any education or socioeconomic status were included. 

### 2.4. Article Screening and Data Extraction

Articles were reviewed in the following stages:Title-screening;Abstract-screening of accepted titles;Full-text screening of accepted abstracts;Data extraction from accepted articles.

Two reviewers conducted all levels of screening and data extraction. Discrepancies were resolved through discussion or review by a third reviewer. 

Data extracted from accepted articles included journal and publication information, participant demographics, inclusion and exclusion criteria, control and intervention descriptions, study duration, outcome measures, results, limitations, and implications (Section A.2). Data collected for framework development comprised details about study design, breath practice, intervention duration, and implementation components, such as use of human-guided training, number of sessions, and use of long-term practice. We defined human-guided training as the use of live or pre-recorded audio or video human guidance during at least the first breath session. Multiple sessions were defined as performing the breathing practice more than once, whether separated by minutes or multiple days. Long-term practice was defined as ≥6 sessions over ≥1 week. 

Authors were contacted when data were unavailable or unclear. 

### 2.5. Assessment of Study Quality

Studies were assessed by two independent reviewers for methodological and reporting quality. Disagreements between reviewers were resolved through discussion or by a third reviewer. RCTs were assessed using a modified form of the Boutron checklist to evaluate a report of a nonpharmacological trial (CLEAR-NPT; Section A.3) [51,62]. Non-RCTs were assessed using the National Heart, Lung, and Blood Institute of the National Institutes of Health (NHLBI-NIH) quality assessment tool for single-arm, pre-post, or non-randomized studies (Section A.4) [63]. For both tools, scores <0.6 were considered poor quality, 0.6–0.8 fair, and >0.80 good. 

All selected studies were included in data extraction and synthesis, regardless of their methodological quality, as assessed with these tools.

### 2.6. Synthesis

Results were synthesized by first classifying the studies by study population, type, and use of stress/anxiety as a primary versus secondary outcome. Patterns among effective and ineffective interventions were identified based on study design and population and breath practice type, training, duration, and implementation. We compared the frequencies of these data to identify factors associated with intervention effectiveness. Parameters were categorized in *n* × 2 contingency tables where rows represented parameters and columns represented effectiveness. Fisher’s exact tests of independence [64,65] were performed for each parameter that could be summarized in a contingency table with fixed margins. Interventions for which effectiveness outcomes differed from the above patterns were labeled outlier interventions, details of which were explored through qualitative analysis to provide further insights into effectiveness patterns. Synthesis results were used to create a conceptual framework of intervention characteristics associated with stress reduction effectiveness. Analyses were performed using R Statistical Software (v4.1.2; R Core Team 2021). Results were presented in narrative, tabular, and figure form.

## 3. Results

### 3.1. Study Selection and Characteristics

Results of the search and screening process are presented in the PRISMA flow diagram (Figure 1).

Table 1 provides details of all included studies by population and by use of stress/anxiety as the primary or secondary outcome. Six broad population categories emerged: youth (4 studies); healthy adults (14 studies); high-anxiety adults (4 studies); clinical populations (29 studies), divided into chronic-care clinical populations (9 studies), such as those with ongoing asthma or hypertension, and acute-care clinical populations (20 studies), such as those undergoing chemotherapy, dental care, or surgery; and individuals placed in simulated-stress situations (7 studies), such as an artificial public speaking task or electric shock anticipation. Forty-one of the 58 studies were RCTs, and 17 were non-randomized prospective clinical studies. Of the 41 RCTs, 23 included psychometric measures of stress or anxiety as primary outcomes, and 18 included these measures as secondary outcomes. Of the seventeen non-RCTs, 8 included psychometric measures of stress or anxiety as primary outcomes, and 9 did as secondary outcomes. Forty-four of the 58 studies were comparative; comparators were usual care or no intervention in 22 studies, other breath-related interventions in 8 studies, and non-breath-related interventions in 14 studies.

Twenty-five of the 58 studies were conducted in Asia; 14 in Europe; 13 in the United States; 1 in Canada; 2 in Brazil; 2 in North Africa; and 1 in New Zealand. Twenty-nine studies were based in hospital or clinical care settings, such as primary care, medical centers, or other clinical settings. Fifteen were at university settings, including medical schools. Three were based at yoga centers, 2 at elementary or secondary schools, 1 in an automotive assembly plant, and 1 was remote. The remaining 6 settings were unclear.

Within the 58 studies, 72 breathing interventions that met our inclusion criteria were identified, and 54 such interventions significantly reduced participants’ stress/anxiety. The most common type of breathing intervention was general slow breathing, which was included in 36 of the studies. The second most common selection was deep diaphragmatic breathing (20), followed by fast breathing (14), alternate or unilateral nostril breathing (ANB, UNB; 11), breathing with holds/pauses (7), extended exhale breathing (6), normal-rate breathing (2), and paced or pursed lip breathing (5). 

The following psychometric measures of stress and anxiety were used (Figure 2): State Trait Anxiety Inventory (STAI) in 25 studies; Visual Analog Scales (VAS) in 7; Beck Anxiety Inventory (BAI) in 6; Hospital Anxiety and Depression Scale (HADS) in 5; stress/anxiety Likert scales in 5; Perceived Stress Scale (PSS) in 4; Profile of Mood States (POMS) in 2; Burn-Specific Pain Anxiety Scale (BSPAS); Depression Anxiety Stress Scales (DASS) in 1; and other stress/anxiety metrics in 11. Additional metrics were included in some studies to evaluate the impact of breathing interventions on cognition, lung function, physiological stress, psychological well-being, and other health-related conditions. 

### 3.2. Study Characteristics and Outcomes by Population

Table A2 describes the stress/anxiety metrics, outcomes, and quality scores of included studies by population.

#### 3.2.1. Youth

Four studies were conducted among youth, of which all were RCTs, and three examined stress or anxiety as a primary outcome. Sample sizes ranged from 27 to 154, mean age from 8 to 15 years, and intervention duration from 1 to 45 days. Of the four interventions in these studies, three used slow breathing and one included a mix of slow breathing, fast breathing, and ANB. All four yielded significant stress/anxiety benefits. 

#### 3.2.2. Healthy Adults

Fourteen studies were conducted among healthy adults, of which eight were RCTs and eight examined stress or anxiety as a primary outcome. Sample sizes ranged from 16 to 761, age from 17 to 70 years, and intervention duration from 1 to 273 days. Of the 21 interventions in these studies, 13 used slow breathing, 1 used regular-paced breathing, 1 used cyclic hyperventilation with holds, 1 used fast pranayama, 1 used slow pranayama, 1 used slow and fast pranayama, 2 used ANB, and 1 included a mix of slow breathing and ANB. Seventeen interventions yielded significant stress/anxiety benefits. 

#### 3.2.3. High-Anxiety Populations

Four studies were conducted among high-anxiety populations, of which two were RCTs and three examined stress or anxiety as a primary outcome. Sample sizes ranged from 18 to 46, mean age from 24 to 43 years, and intervention duration from 2 to 56 days. All four interventions in these studies incorporated some type of deep, diaphragmatic and/or slow breathing, and all yielded significant stress/anxiety benefits. 

#### 3.2.4. Clinical Populations, Acute

Of the 20 studies conducted in clinical settings among populations with acute clinical conditions, 15 were RCTs and 16 examined stress or anxiety as a primary outcome. Sample sizes ranged from 20 to 272, mean age from 21 to 75 years, and intervention duration from 1 to 56 days. Of the 21 interventions in these studies, one used ANB, three included a mix of slow and fast breathing, and the rest used slow breathing. Seventeen interventions yielded significant stress/anxiety benefits. 

#### 3.2.5. Clinical Populations, Chronic

Of the nine studies conducted in clinical settings among populations with chronic health conditions, six were RCTs and three examined stress or anxiety as a primary outcome. Sample sizes ranged from 11 to 655, mean age from 31 to 67 years, and intervention duration from 1 to 182 days. Of the eleven interventions used within these studies, six used slow breathing, one used fast breathing, one used UNB, and three included a mix of fast and slow breathing. Seven interventions yielded significant stress/anxiety benefits. 

#### 3.2.6. Simulated Stress Settings

Of the seven studies conducted among populations undergoing simulated stress situations, five were RCTs and three examined stress or anxiety as a primary outcome. Sample sizes ranged from 30 to 111, mean age ranged from 23 to 28 years among studies reporting mean data and from 19 to 24 among studies not reporting means, and intervention duration for all was one day. Of the eleven interventions in these studies, six used slow breathing, one used regular-paced breathing, one used fast breathing, one used fast pranayama, and two used ANB. Five interventions yielded significant stress/anxiety benefits. 

### 3.3. Study and Reporting Quality

Of the 58 included studies, 7 RCTs and 0 non-RCTs received “good” quality ratings, 20 RCTs and 9 non-RCTs received “fair” ratings, and 14 RCTs and 8 non-RCTs received “poor” ratings (Table A2). Figure 3 summarizes the top identified quality issues. Among RCTs, the main concerns related to the absence of the following: blinding, intention-to-treat analysis, concealed treatment allocation, and analytic consideration of between-group differences in lost-to-follow-up and withdrawals. Among non-RCTs, the main concerns related to the absence of the following: repeated measurements, assessor blinding, population inclusiveness, adequate sample size, and intervention description in the report. The reviewers noted other common quality issues during abstraction. Many pre-post studies did not include a control group, and those that did lacked an active control. Several intervention descriptions were vague and lacked detail about factors such as breath rate, bpm, and instructions for breath practices. Adherence and compliance with at-home practice were measured and reported poorly or not at all, and reports on human-guided training lacked detail in duration, timeline, and extent of guidance. 

### 3.4. Synthesis Results

#### 3.4.1. Study Design Parameters

The following study design parameters were identified for analysis regarding their association with intervention effectiveness: study type (RCT, non-RCT); stress/anxiety as the primary or secondary outcome; control group type (usual care/no intervention, breath practice + other components, such as yoga or meditation, non-breath active control, meditation or mindfulness, no control); and the number of stress/anxiety outcome measurements (1, 2, 3+).

None of the study design parameters were significantly associated with intervention effectiveness (Table 2). RCTs made up 72% of effective (39 of 54) and 67% of ineffective (12 of 18) interventions, with these proportions not differing significantly (*p* = 0.78). Stress/anxiety was a primary outcome in 67% of effective and 61% of ineffective interventions (*p* = 0.67). There was no association between effectiveness and use of control group (*p* = 0.19). Stress/anxiety was measured as follows: once in 4% (2 of 54) of effective and 6% (1 of 18) of ineffective interventions; twice, once before and once after the intervention, in 54% and 67% of the respective interventions; and three or more times in 43% and 28%, respectively (*p* = 0.43). 

#### 3.4.2. Breathing Practice and Implementation Parameters 

The following breath practice and intervention implementation parameters were identified for analysis: population category; type and pace of breath practice (slow-only, fast-only, a combination of fast and slow with or without ANB/UNB, regular pace, ANB/UNB alone); the use of human-guided training (yes/no); number of human-guided sessions (0, 1, 2–3, 4+); among device-guided interventions, use of human-guided training (yes/no); number of sessions (single or multiple); long-term practice (≥6 sessions over ≥1 week, yes/no); intervention duration (1, 2–5, 6–30, or 31+ days); session duration (<5, 5–10, 11–20, or >20 min); group- or individual-only or group + individual practice; and inclusion of at-home individual practice (yes/no). 

Intervention effectiveness was not significantly associated with population category (*p* = 0.14), session duration beyond 5 min (*p* = 0.81), group versus individual versus group + individual practice (*p* = 0.79), and use of at-home practice (*p* = 0.24; Table 3). Among youth and high-anxiety adults, all interventions were effective. Among healthy adults, 17 of 21 (81%) interventions were effective; these numbers were 17 of 21 (81%) among acute clinical populations, 7 of 11 (64%) among chronic clinical populations, and 5 of 11 (45%) among simulated stress populations. The majority of interventions were individual-only (61% of effective and 56% of ineffective interventions) or group-only (28% and 33%, respectively), with 6% of effective and 0% of ineffective interventions including both group and individual sessions. Five interventions were unclear about this parameter in the published paper. Session durations ranged from 1.3 to 60 min for effective and 1–45 min for ineffective interventions. Six percent of effective and 33% of ineffective intervention session durations were <5 min, and 80% of effective and 50% of ineffective interventions’ sessions lasted ≥5 min; duration was unspecified in the remaining 11 interventions. Among 5 min or longer interventions, 30% of effective and 17% of ineffective ones were 5–10 min, 19% of effective and 17% of ineffective ones were 11–20 min, and 31% of effective and 17% of ineffective ones were >20 min. Six percent of effective and zero ineffective interventions were combined group and individual, and 6% of effective and 11% of ineffective interventions did not report group versus individual implementation. At-home practice was included in 35% of effective and 17% of ineffective interventions. 

Intervention effectiveness was significantly associated with the following parameters: breath pace (*p* = 0.01); use of human-guided training (*p* < 0.001); number of human-guided sessions (*p* < 0.001); among device-guided interventions, use of human-guided training (*p* = 0.025); number of sessions (*p* < 0.001); long-term practice (*p* = 0.005); intervention duration (*p* < 0.001); and session duration <5 versus ≥5 min (*p* = 0.005). Effectiveness was no longer associated with breath pace when considering any pace besides fast-only (*p* = 0.10) or when comparing fast-only to any other pace (*p* = 0.06). All three human-guided training parameters were subsumed into the broader category of human-guided training, as were all three intervention duration measures into long-term practice. Thus, five main parameters were identified as components associated with stress reduction effectiveness, which were subdivided into exclusionary and core components. Breath pace and session duration comprised exclusionary components, in which avoiding fast-only practices and sessions <5 min appears conducive to but not sufficient for intervention effectiveness, while incorporating the core components of human-guided training, multiple sessions, and long-term practice actively contributes to stress reduction benefits. 

Figure 4 shows the numbers of effective and ineffective interventions by breath pace. Figure 5 shows how effective and ineffective interventions were distributed by population (Figure 5A) and by use of human-guided training, multiple sessions, and long-term practice, both overall (Figure 5B–D) and by population (Figure 5E–G). 

Overall, 44 of the 54 effective and 10 of the 18 ineffective interventions incorporated some form of human-guided training during participants’ initial breath practice session (Figure 5B), with a total number of human-guided sessions ranging from 1 to 40. These sessions ranged from 3 to 60 min among effective interventions and from 3 to 45 min among ineffective ones. Fifty of the 54 effective and 1 of the 18 ineffective interventions incorporated multiple breath practice sessions throughout the intervention (Figure 5C), with up to 100+ sessions incorporated. Long-term practice was incorporated in 27 of the effective and 2 of the ineffective interventions (Figure 5D), ranging from 1 week to 3 months of practice.

Considering each population specifically, interventions in youth and high-anxiety adults were always effective (Figure 5A,E). All eight of these interventions incorporated at least one of the three core components of human-guided training, multiple sessions, and long-term practice; and none used fast-only or <5 min practices. In healthy adults, all seventeen effective interventions incorporated at least one core component, and all four ineffective ones used zero core components and <5 min sessions. The effective interventions in clinical and simulated-stress populations always used at least one core component; three used <5 min sessions. Among the fourteen ineffective interventions within those populations, seven used one or more core components; of these seven, one used <5 min sessions, and another used both fast-only and <5 min practices. 

Synthesis outliers were defined as effective interventions that did not incorporate any of the three core components or ineffective interventions that avoided both fast-only breathing and sessions <5 min and incorporated at least one core component. Six outlier interventions were identified and explored to determine explanatory variables that may be considered potential model caveats (Table 3). One intervention was effective despite not incorporating human-guided training, multiple sessions, or long-term practice [79]. Effectiveness may relate to the investigator’s presence throughout the device-guided practice, which may have led to unplanned human guidance that was not reflected in the study report. It is also possible that the highly educated and physiologically trained 5th-year medical residents who participated in this study easily adapted to learning a breath practice. 

Five interventions were ineffective despite avoiding fast-only and <5 min practices and incorporating at least one core component (Table 3). In Meier (2020), the fact that participants were standing still for a 30 min practice and were repeatedly interrupted for stress surveys likely explains the lack of effectiveness. The ANB intervention in Kamath (2017) was ineffective despite incorporating both human-guided training and multiple sessions. Of the three other ANB interventions used in simulated stress situations, the one effective intervention [107] incorporated human-guided training among a population with prior yoga breathing experience [107]. The two ineffective ones had either human-guided training in a yoga-naive population [44] or no such training in a yoga-breathing-experienced population [49]. All other ANB studies were effective in reducing stress and incorporated ≥2 of the three core components. It may be that in a highly technical practice such as ANB, multiple sessions or prior experience plus human-guided training are required for stress-reducing effects. 

In Ratcliff (2019), effectiveness may have been diluted by the difficulty of doing diaphragmatic breathing while lying still and face-down on a procedure table during a breast biopsy. Investigator bias may also have influenced effectiveness, with unblinded investigators leading this active control practice. In Thomas (2017), the practice was not continued by many participants for the entire 12-month measurement window.

#### 3.4.3. Framework Development

Synthesis results were used to develop a conceptual framework of components associated with effective stress reduction breathing interventions (Figure 6): avoiding fast-only breathing and session durations <5 min; and including human-guided training, multiple sessions, and/or long-term practice. The framework also identifies caveats reflecting factors that may potentially reduce intervention effectiveness, such as extensive standing, repeated interruptions, involuntarily restricted diaphragmatic expansion, and inadequate training for highly technical practices. Intervention effectiveness may also be impeded in studies that lack blinded practice guides, accountability for at-home practices, and practice continued throughout measurement periods.

## 4. Discussion

### 4.1. Overview

We reviewed the peer-reviewed published literature to evaluate practice components associated with stress reduction effectiveness. We identified 58 clinical studies, comprising 41 RCTs and 17 single-arm, pre-post, non-randomized trials; 5407 total participants were included, with a mean age of 35.9 years and 43.92% female. Among the 72 included interventions, breathing practices significantly reduced stress/anxiety in 54 interventions. Findings were used to develop a conceptual framework of breath practice effectiveness, specifying that stress-reducing interventions incorporated five components: avoidance of fast-only breathing and sessions <5 min; and inclusion of human-guided training, multiple sessions, and/or long-term practice. Neither the population doing the practice, other breath paces besides fast-only, session duration beyond 5 min, group versus individual practice, nor setting were associated with effectiveness. 

Breathing practices have become an increasingly popular stress reduction tool with widespread benefits for mental health. A simple PubMed search of “breathing practices” yields 18,721 results, over half of which have been published since 2010. Mobile applications are bringing breathwork to broader audiences, with the number of existing breath apps in the double digits. Despite this popularity, not everyone is aware of the benefits of breathwork; among those who are, emphasis is often placed on specific practice types and durations. Though many breath coaches pride themselves in providing the “best” practice, no singular effective protocol has been established and supported by scientific analyses. 

Previous reviews have provided insight into the psychological and/or physiological benefits of slow breathing [52], diaphragmatic breathing [36], ANB [38], and other breath practices [23,25,35,37]. While these studies have examined a range of populations, their selection criteria have precluded more nuanced analyses. Although the analysis by Fincham and colleagues attempted to investigate subgroup effects, they included only 12 studies [35]. In addition, most reviews include breathing interventions incorporating biofeedback [23,25,35,52], which is not readily applicable to at-home practice. Ours is the first work of its kind to disentangle among varied populations the components of isolated breathing interventions that contribute to stress and anxiety reduction from those that do not. The framework provides clear, specific recommendations for individuals implementing their own practices, professionals developing and leading stress reduction programs, and researchers studying the effects of breath practices on stress outcomes. 

### 4.2. Exclusionary Components

#### 4.2.1. Breath Pace 

Intervention effectiveness was less likely during fast-only breathing versus any other pace, consistent with other reviews finding benefits associated with slow-paced breathing practices [23,35]. Polyvagal [108] and neurovisceral integration [109] theories favor slow over fast breathing due to its activation of the parasympathetic nervous system and accompanying effects on the mind and brain [52]. Our analysis further suggests that, provided fast-only practices are avoided, specific breath pace may not matter, as effectiveness was not associated with pace when fast-only practices were excluded. 

Interestingly, fast breathing can successfully reduce stress when paired with slower paces or holds, as seen in the 12 fast + slow interventions, all of which were effective. Such practices may provide benefits by teaching practitioners to slow their breathing patterns and self-soothe during stressful situations; or by building CO_2_ tolerance, which is inversely related to symptoms of stress and anxiety [110]. They may, alternatively, train the autonomic nervous system to engage the parasympathetic branch more readily following high sympathetic activation [111]. Yet another interpretation resides in the connection between breathing patterns and the brain’s neural circuitry, whereby shifting breathing patterns may correspondingly shift neural circuit dynamics and thus mood and mental state [26]. 

If this explanation holds true, regular-paced breathing practices would provide no such benefit. Because only two interventions met our 12–20 bpm regular-paced breathing definition, we cannot definitively assess the stress-reducing effects of regulated breathing at one’s normal pace. Of the two regular-paced interventions, one was ineffective, which may relate to its lack of framework adherence [42]. The other was effective, possibly due to human-guided training and its 12 bpm pace, which may have been slow enough to engender benefits [78].

The lack of effectiveness observed among fast-only breathing interventions should be interpreted with caution, given their underrepresentation in our analyses. The two fast-only interventions either lacked framework adherence [48] or incorporated poor study design [105]. Should breathwork’s benefits relate to respiration-induced neural entrainment, any breathing pace different from one’s usual pattern may offer stress relief—even fast-only practices. Fast breathing could hypothetically reduce stress by acting as an acute stressor that builds resilience over time or by providing a sense of control from learning to regulate one’s own breathing and internal state. 

#### 4.2.2. Session Duration 

The significant association between intervention effectiveness and session duration was driven by interventions using sessions <5 min, six of nine of which were ineffective. Among these, however, four incorporated no core component, and the remaining two included other factors that may have influenced effectiveness. Regarding the latter, stress reduction may have also been hindered by fast-only breathing in Watson and colleague’s intervention and by the discomfort of doing deep diaphragmatic breathing after abdominal surgery in Aktas and colleagues’ intervention [39,105]. The presence of core components in the three effective interventions [39,105,106] suggests that such short sessions may be particularly sensitive to practice implementation, such that human guidance, multiple sessions, or long-term practice are required for effective stress reduction.

The lack of significant association between effectiveness and session durations ≥5 min indicates that any session duration beyond 5 min can be effective. While no other studies have assessed differences in breath practice effectiveness due to session length, many generally assume a dose-response relationship in which longer sessions are inherently better. Our findings indicate that simply engaging in a breathing practice provides benefits, with sessions as short as 5 min yielding comparable benefits to longer sessions. Interestingly, a recent review by Strohmaier and colleagues reported similar findings about mindfulness-based programs, with larger doses of mindfulness practices no more effective than smaller ones [112]. 

### 4.3. Core Components

#### 4.3.1. Human-Guided Training

The association between human-guided training during initial sessions and effectiveness highlights the value of proper breath mechanics, diaphragmatic engagement, and the ability to control breath pace. Hearing real-time instructions allows participants to receive performance cues and, in the case of live guidance, to ask questions and/or get feedback. Such guidance may enhance adherence and stress reduction outcomes. The fact that this training could be live or pre-recorded and as minimal as one 5 min session implies accessibility to broad populations, although it is notable that all effective interventions among high-anxiety populations included two or more human-guided sessions. 

While important, human-guided training does not guarantee stress reduction benefits, and the lack thereof does not prevent effectiveness. Effectiveness may be compromised despite human guidance in the presence of clinical-related breathing restrictions, fast-only or highly technical breathing techniques, and prolonged latency between intervention and outcome measurements. More human guidance may be needed in situations of clinical stress, such as undergoing a breast biopsy [47], for learning more technical practices, such as ANB and UNB, or for high-anxiety populations. In some cases, lack of human-guided training may be made up for by multiple sessions that allow participants to learn these nuances themselves over time. Of the four effective interventions that lacked such training, three included multiple sessions [39,80,106], and the other [79] may have incorporated some unreported and unintentional human guidance due to the investigator’s presence during the practice.

#### 4.3.2. Multiple Sessions

The observed importance of multiple sessions likely relates to greater practice familiarity and consistency. Multiple sessions took various forms, ranging from two guided individual sessions in one day [67] to five group sessions over 9 months alongside twice-daily at-home practice [71]. Intervention effectiveness persisted despite such heterogeneity. 

Although 41 of the 44 effective multi-session interventions included at least one human-guided session, having multiple sessions alone appears to support breath practice effectiveness. Three of the four effective unguided interventions included multiple sessions [39,80,106]; and significant associations between effectiveness, intervention duration, and number of sessions suggest that breathing practices are more beneficial when performed repeatedly. Nevertheless, some single-session interventions were, in fact, effective, with 10 of 23 providing stress reduction benefits. Nine of these 10 were human-guided, suggesting that adequate training may, in some cases, make up for multiple sessions in breath practice efficacy. 

Examining the four multi-session interventions that lacked human guidance also suggests that interventions without human-guided training are more likely to be effective when they include simple, easy-to-follow breathing practices alongside device guidance. Interestingly, seven of eight single-session interventions with such simple, device-guided practices were ineffective, highlighting the importance of multiple sessions when human guidance is lacking.

#### 4.3.3. Long-Term Practice

Practicing for a minimum of six sessions over at least a week was significantly associated with effectiveness, suggesting that breathing practices are more likely to confer stress-reducing benefits when performed regularly over time. As with many behaviors, consistent practice is often key to effective skill learning and adaptation. Although these results contrast with other findings of no treatment duration effect [24,35], those reviews may have been limited by small samples, study heterogeneity, and exclusion of human-guided training, which may interact with long-term practice.

All long-term interventions in our review also incorporated human-guided training, suggesting that human-guided training may underlie the association we found between long-term practice and effectiveness. However, examining the relationship further suggests otherwise: 6 of the 27 effective interventions with long-term practice had only one human-guided session, with the remaining sessions performed at home, unguided. If human-guided training were solely responsible for intervention effectiveness, these interventions likely would not have demonstrated benefits, as their outcome measures were taken up to 45 days after human-guided training [20]. It is more likely that long-term practice enhances intervention effectiveness when coupled with at least one human-guided session, which may help teach novice practitioners proper technique.

### 4.4. Components Not Associated with Effectiveness

Our analysis found no association between effectiveness and study population. Because breathing acts directly on the autonomic nervous system as a “bottom-up” approach to stress reduction [113,114,115,116,117,118], its benefits likely persist across diverse populations. This finding contrasts with those from Fincham and colleagues, whose review reported a significant benefit of breathwork on stress in non-clinical populations only [35]. However, their analysis may have been underpowered with only one intervention in mental health populations; and another recent meta-analysis found significant effects in both psychiatric and healthy populations [24]. 

It Is worth noting that all interventions in youth and high-anxiety populations were effective. Perhaps this is a coincidence due to the small sample sizes of four interventions each or due to the prevalence of human-guided training and multiple sessions in these interventions. These two core components may alone be a golden ticket to effective stress reduction with breathing practices. Of the 45 interventions in any population that incorporated both of these components, only four were ineffective [44,47,50], but this was likely due to taxing clinical situations [47], inadequate training or experience for a more technical practice [44], and practice discontinuation [50]. Alternatively, the effects of breathwork may be more potent in high-anxiety populations and youth. The respiratory dysregulation and high chronic stress accompanying anxiety disorders may augment sensitivity to breath practice interventions [53]. Accustomed to being in learning environments, youth may easily catch on to these practices with human guidance, and their developing nervous systems may be highly receptive to interventions [119]. 

We found that interventions given to groups were equally effective as those given to individuals, a finding corroborated by other recent literature [35]. However, given the small number of group interventions in high-anxiety and acute chronic populations, it is possible that populations with particularly dysregulated breathing [28,29,33] may fare better with one-on-one attention. 

Interventions with multiple at-home practice sessions were equally as effective as those with multiple sessions administered via human guidance. Thus, human-guided training does not appear necessary beyond the first session, after which breathing practices can be carried out individually at home. Nonetheless, interventions incorporating at-home practices should motivate and assess practice compliance, as our three ineffective interventions with at-home practice [39,50] reported either inconsistent or unknown practice compliance. 

### 4.5. Framework Caveats

By scrutinizing the six outlier interventions, we identified framework caveats that may either improve or undermine effectiveness, independent of the proposed framework components. Regarding the former, our analysis indicates that even breathing practices that diverge slightly more from one’s spontaneous pattern may not require human-guided training, given adequate device guidance paired with strong accountability and physiologically trained participants. While Schlatter and colleagues’ effective intervention [79] lacked human guidance, their 6 bpm practice among 5th-year medical residents was guided via a screen cursor with an investigator present to ensure practice adherence.

On the physical side, breath practices should be performed in a comfortable position without prolonged standing or involuntary diaphragmatic immobilization. The 30 min of standing during Meier and colleagues’ human-guided intervention may have caused discomfort and interfered with potential benefits [46]; though perhaps shorter standing practice durations would be effective. Deep or diaphragmatic breathing may provide minimal benefit when voluntary full diaphragmatic movement is lacking, such as during a breast biopsy [47]. Practices placing less emphasis on the diaphragm and more emphasis on breath cadence, such as extended exhale or general slow breathing, may offer better relief in such situations. 

Breath practices may also be less likely to reduce stress when they include interruptions. While practicing breathing in a standing position may have influenced Meier’s findings, so could its repeated data collection interruptions, which may have reduced focus and induced sufficient stress to negate benefits. 

Analysis of included ANB/UNB interventions suggests that more technical practices may require extra training or experience. One ANB intervention was ineffective despite fitting our framework, incorporating a single human-guided session among yoga-naive participants [44]. Of our four effective ANB/UNB interventions carried out in novice practitioners, all included human-guided training for a minimum of four sessions [69,77,88,100]. These findings are consistent with ANB’s stress-reducing effects reported elsewhere, in which all included interventions incorporated long-term practice alongside human-guided training [38]. Nonetheless, an effective human-guided but short-term intervention among individuals with prior ANB or yoga breathing experience suggests that single-session ANB may confer benefits when carried out among experienced practitioners and with human-guided training [107]. The technical nature of learning and practicing ANB may initially attenuate its stress-reducing effects, specifically for inexperienced practitioners, until repeated practice occurs. Conversely, human-guided training and long-term practice may not be necessary for exceedingly simple breathing practices, such as those at a slightly slower-than-normal pace (e.g., 10 or 8 bpm) that may be more accessible to breathwork-naive individuals [80,106]. 

Our findings also suggest that long-term effectiveness requires long-term compliance. Participants in Thomas and colleagues’ 2017 ineffective interventions engaged in regular practice for 6 to 8 weeks but were not practicing most days by the time of the 12-month stress measurement [50]. Another review found that breathwork benefits may persist for a brief period after practice ends, although practice adherence was not assessed in that study [35]. In our analysis, all other studies with long-term measurements included either regular guided practice throughout or specific instructions and motivation for at-home practice. 

### 4.6. Limitations and Future Directions

Findings from this analysis must be considered in light of its potential limitations. Due to the mediocre-at-best methodological quality of included studies, some effectiveness findings may not be as reliable or valid as assumed. Future research should reduce potential bias by incorporating appropriate blinding and measurements, intention-to-treat analyses, concealed treatment allocation, and larger sample sizes. The articles included in our analysis demonstrated wide heterogeneity in reporting practice and implementation details, often omitting key data relevant to our framework. Zaccaro and colleagues propose a checklist of data points to include in breath studies [52], including factors such as breath pace, inspiration/expiration ratio, pauses/holds duration, and others. Studies should also report on implementation components, such as human-guided training, number of sessions, practice compliance, session durations, group-versus-individual practice, etc. In the comparative studies included in this review, lack of active controls may have biased results with placebo effects. Future research should compare breath practices with other self-care interventions that are framed as helping with stress reduction. 

Although we aimed to address breathing practices performed in isolation and accessible to broad populations, results may have differed if we had included those combined with other practices such as yoga, meditation, or biofeedback. For example, HRV is a well-established stress measure [120]. Changes in self-reported stress may differ when participants see HRV responses and modify their breath pace to achieve certain HRV outcomes. More research is needed comparing biofeedback-enabled breath practices with breath practices alone to help disentangle the cause-and-effect relationships between breath practices, HRV, and stress. Physiological outcomes were also not considered in our analysis. While these outcomes are helpful for understanding breathing practices’ mechanisms of action, our goal was to develop a framework for practical application, regardless of the mechanism of action. 

The framework presented here is based on associations between breath practice components and effectiveness; it does not necessarily represent causal relationships. A meta-analysis that pools all data into one larger analysis was not conducted due to heterogeneity in populations, breath practice details, outcome metrics, and statistical approaches in included studies. While existing meta-analyses provide valuable insights [24,35,121], we included a broader array of studies to establish a deeper understanding of how various practice and implementation factors are associated with stress reduction. 

Although our results suggest that any breath pace besides fast-only can be effective when the framework criteria are applied, there may be finer distinctions among populations and specific outcomes of interest in which different practices help specific populations in different ways. More carefully controlled and methodologically rigorous experiments may reveal whether these more nuanced distinctions exist.

The potential caveats identified in this framework are hypotheses based on data available in outlier interventions; it could be that other factors drove effectiveness in these interventions. Only two databases were searched for this systematic review; however, the large number of identified and included studies lends credence to the conclusions presented. Lastly, caution is warranted in applying these results to individuals with certain clinical conditions, such as epilepsy, heart conditions, or pregnancy, whose reactions to intense practices may vary. 

### 4.7. Implications

We have identified simple yet important criteria for learning and implementing breath practices for effective stress reduction. Our research demonstrates that the important factors to consider when implementing breath practices for stress reduction are avoiding fast-only and <5 min practices and incorporating appropriate training and continued practice. Extensive standing, interruptions, involuntarily restricted diaphragmatic movement, and inadequate training for highly technical practices may render otherwise promising interventions ineffective. Besides these factors, practitioners and program developers can be less concerned with the specific details of a breath practice, such as finer distinctions in pacing, durations beyond 5 min, group or individual practice, and setting. While these still need to be considered in implementing any breath practice, our findings indicate that over-emphasis on these details is unwarranted and a misdirected placement of focus. 

This knowledge can and should provide a sense of freedom for individual practitioners and program developers alike in tailoring programs to meet their needs for stress reduction effectiveness. While many people may benefit from a simple breath practice, some may be reluctant to start for fear of doing the “wrong method.” Our results not only support the broad-ranging benefits of a variety of simple breathing techniques but also provide comfort in the knowledge that, as long as this simple framework is followed, other details are less important for maximizing stress reduction benefits of breathing practices. Specific practices to consider include extended exhale breathing, box breathing, slow breathing with pauses/holds, slow diaphragmatic breathing, combined fast + slower breathing practices, and ANB/UNB. Both initial training sessions and ongoing practice sessions can be as short as 5 min. Breath practice programs should incorporate initial human-guided training and be clear about the specific long-term practice criteria for success.

Future breath practice studies should incorporate rigorous data collection and report breath practice implementation details to allow for analysis of finer practice distinctions and this framework’s validity. Real-time guidance in these studies should be carried out by individuals blinded to experimental conditions, and practices should continue throughout the measurement period. Accountability for, motivation for, and assessment of practice compliance should be addressed, especially for home practices and longer-term follow-up periods. Until more such research is conducted, following the framework results from this analysis will improve stress management effects associated with breathing practices. 

### 4.8. Actionable Takeaways

Following the below framework criteria will increase the likelihood of beneficial stress outcomes associated with breath practices (Table 4). 

## 5. Conclusions

We developed an evidence-based framework for effectively implementing stress reduction breath practices. Despite prior knowledge that breathing practices support stress reduction, how these effects vary across populations, breath practice types, and implementation approaches remained unclear. While commonly emphasized factors such as specific breath pace, population, and group versus individual practice were not found to be associated with effectiveness, five core components were: avoiding fast-only and <5 min practices; and incorporating human-guided training during initial sessions, multiple sessions, and long-term practice. Following this simple, evidence-based framework can help maximize the stress reduction benefits of breathing practices among broad populations. 

## Figures and Tables

**Figure 1 brainsci-13-01612-f001:**
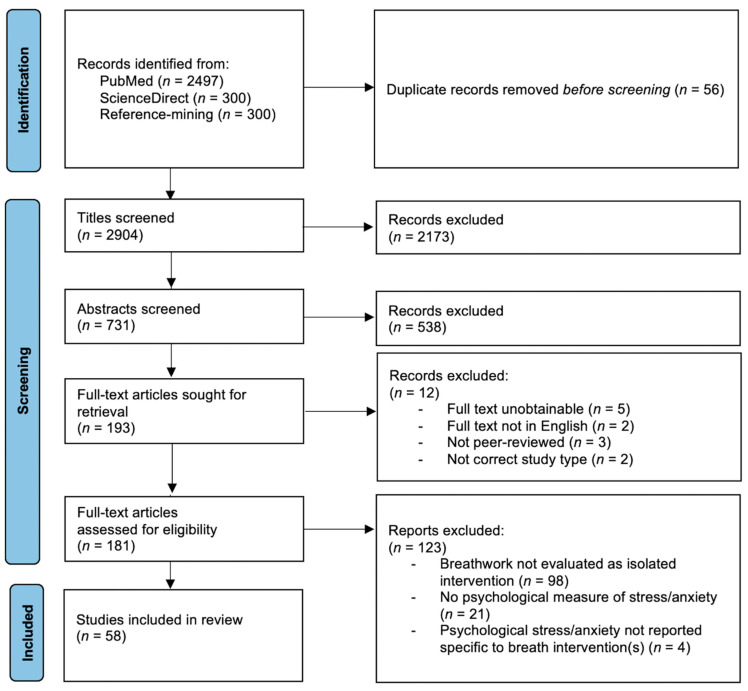
Preferred Reporting Items for Systematic Reviews and Meta-Analyses (PRISMA) flow diagram of study identification, screening, and selection process. *n*, number.

**Figure 2 brainsci-13-01612-f002:**
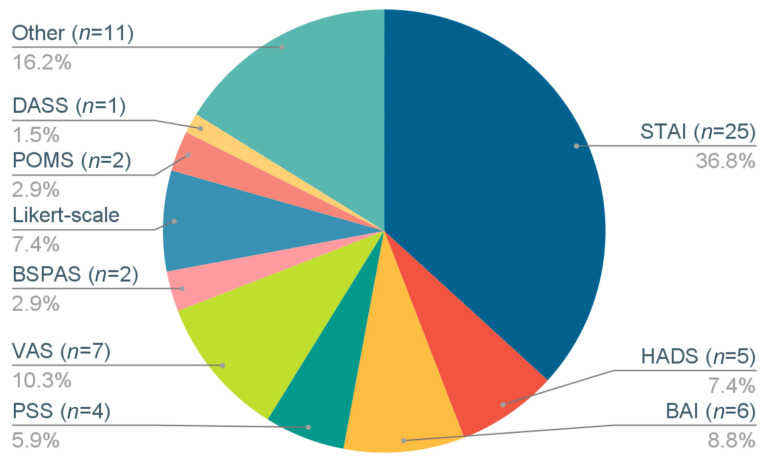
Distribution of stress/anxiety outcome metrics used in included studies. Rounding errors result in whole-number percentages summing to more than 100%. Total number of studies using metrics sums to more than 58 because some studies incorporated more than one outcome metric. STAI, State-Trait Anxiety Inventory; HADS, Hospital Anxiety and Depression Scale; BAI, Beck Anxiety Inventory; PSS, Perceived Stress Scale; POMS, Profile of Mood States; BSPAS, Burn-Specific Pain Anxiety Scale; DASS, Depression Anxiety Stress Scale; VAS, Visual Analog Scale.

**Figure 3 brainsci-13-01612-f003:**
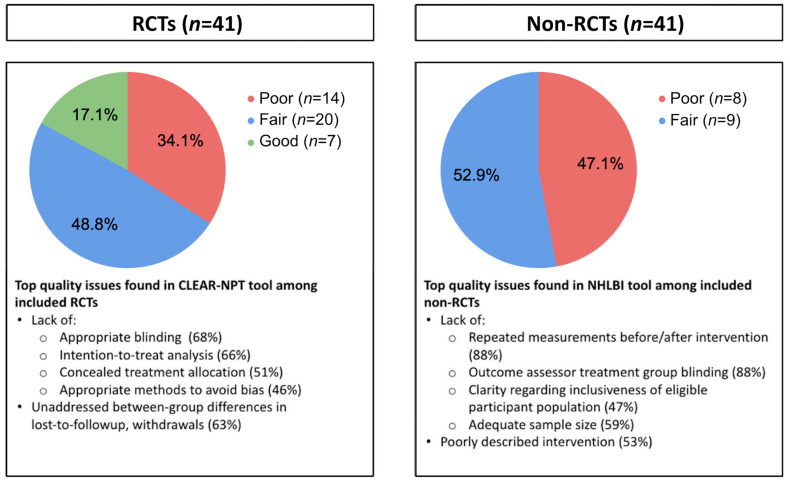
Top quality issues in 58 studies included in review by study type. Quality score calculated by dividing number of met criteria by number of applied criteria and normalizing to 0–1.0 range. Quality score categories defined as follows: <0.6, poor; 0.6–0.8, fair; and >0.80, good. CLEAR-NPT, checklist to evaluate a report of a nonpharmacological trial; NHLBI, National Heart, Lung, and Blood Institute; RCT, randomized controlled trial.

**Figure 4 brainsci-13-01612-f004:**
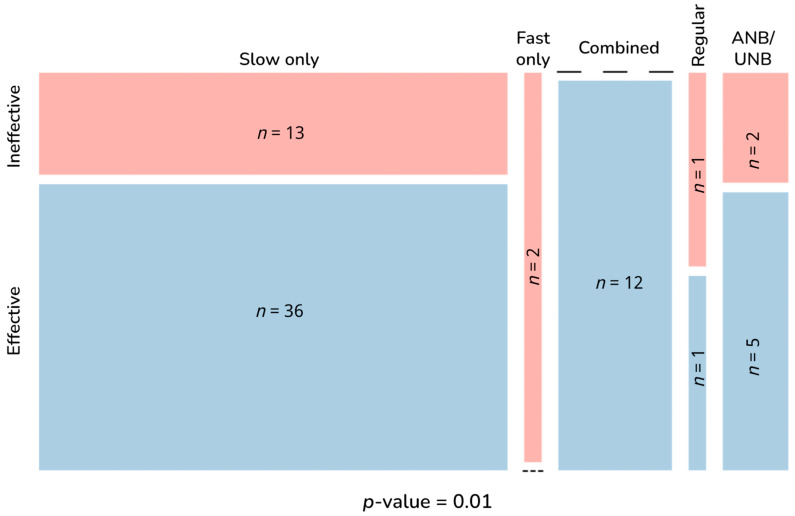
Effective and ineffective interventions by breath practice pace. Blue shaded portion in each category column represents the proportion of effective interventions; red shaded portions reflect ineffective interventions. Dashes represent the lack of effective interventions in the fast only category and ineffective interventions in the combined category. *p*-value calculated with Fisher’s exact test. Breath intervention types defined as fast, breath pace >20 bpm; slow, <12 bpm; regular, 12–20 bpm; combined, fast, and slow breathing with or without ANB/UNB; and ANB/UNB. ANB, alternate-nostril breathing; *n*, number of interventions; UNB, unilateral nostril breathing.

**Figure 5 brainsci-13-01612-f005:**
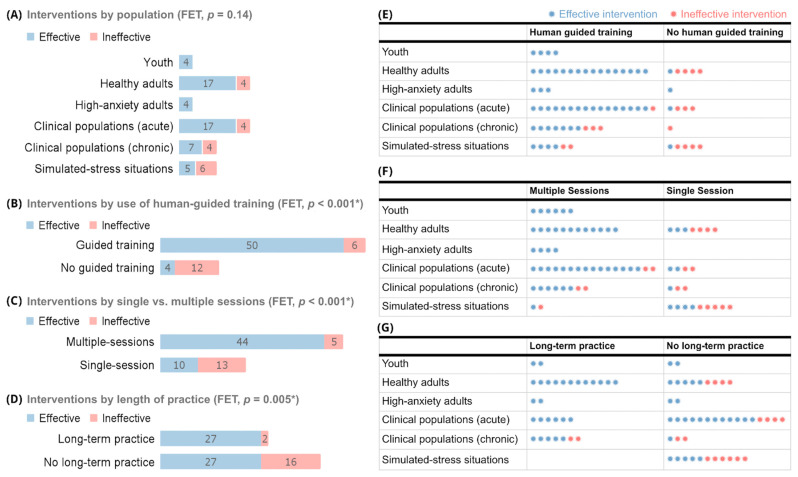
(**A**–**G**) Effective and ineffective breath interventions by population and use of human-guided training, single or multiple sessions, and long-term practice. Each panel is collectively exhaustive of all 72 interventions from 58 studies in total. Numbers in bar graphs of panels (**A**–**D**) represent numbers of interventions. Each colored asterisk in panels (**E**–**G**) represents one study. Blue bars and asterisks represent effective interventions; pink ones represent ineffective interventions. Human-guided training defined as use of live or pre-recorded audio or video human instruction throughout at least the initial breath session. Multiple sessions defined as performing breathing practice more than once. Long-term practice defined as performing ≥6 sessions over ≥1 week. (**A**,**E**–**G**) Interventions were divided into six broad population categories: youth (n = 4); healthy adults (n = 21); high-anxiety adults (n = 4); clinical, chronic (n = 11); clinical, acute (n = 21); and individuals placed in simulated-stress situations (n = 11). FET, Fisher’s Exact Test. * Statistically significant at *p* < 0.05.

**Figure 6 brainsci-13-01612-f006:**
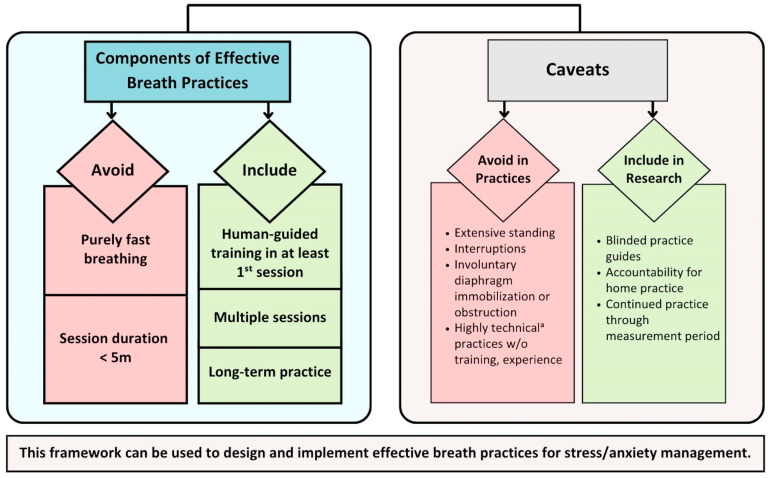
Conceptual framework of effective stress reduction breath practices. Based on analysis of 58 studies with 72 total interventions. Components in left-hand box were significantly associated with effectiveness. Caveats describe factors that may impact intervention effectiveness regardless of component inclusion. Human-guided training defined as use of live or pre-recorded audio or video human instruction throughout at least the initial breath session. Multiple sessions defined as performing breathing practice more than once. Long-term practice defined as performing ≥6 sessions over ≥1 week. ^a^ Highly technical includes breath pace far from regular pace, e.g., extremely fast or extremely slow, or ANB, UNB. ANB, alternate-nostril breathing; UNB, unilateral nostril breathing; w/o, without.

**Table 1 brainsci-13-01612-t001:** Details of included studies’ design and interventions (N = 58) by population and use of stress/anxiety as primary or secondary outcome.

Author, Year, Setting	Study Design, Population	Mean Age (SD), % Female, N	Intervention and Control, Time Frame
**A. Youth**			
Youth: Stress/anxiety as primary outcome			
* **Sellakumar, 2015** [20] India, secondary school (grades 6–12)	RCT Adolescent school students	14 y, 48% Ix: 50 C: 50	Ix: 1-time guided group training followed by 45 days of 30 min individual: **rhythmic breathing, pranayama, SK, slow breathing, deep breathing, prolonged exhales, brief breath-holds** C: No intervention
* **Khng, 2017** [66] Singapore, 4 elementary schools	RCT 5th-grade students	Ix: 10.7 y, 48%, 61 C: 10.7 y, 48%, 61	For 10 min prior to and during six 1 min breaks in 2nd administration of math test: Ix: **Deep abdominal breathing**; C: Resting quietly
* **Bargale, 2021** [67] India, hospital pediatric dentistry dept.	RCT Children needing dental treatment under anesthesia	51% Ix: 8.5 y (1.9), 30 C: 8.4 y (1.9), 30	Two sessions, before and after anesthetic procedure, of guided: Ix: Slow inhale, 2 s hold, exhale into pinwheel C: **Relaxation breathing,** feeling chest and belly move
Youth: Stress/anxiety as secondary outcome			
* **Hakked, 2017 [68]** India, swim academy	RCT Competitive youth swimmers	52% Ix: 15.2 y, 14 C: 15.1 y, 13	For 4 wks, regular swim training alongside: Ix: 10 m, 5×/wk in group w/audio-guided yogic breathing practices: * **Sectional (abdominal, thoracic, upper lobar) breathing** * Yogic **bellows breathing** (forceful nasal breathing w/o strain and w/abdominal expansion/contraction) * **ANB w/breath retention** after inhaling C: No intervention
**B. Healthy Adults**			
Healthy Adults: Stress/anxiety as primary outcome			
* **Gupta, 2010 [69]** India, yoga training camp	Pre-post trial Married men	60–70 y (range), 0%, 30	7-day group training + 3 mo individual practice of **ANB**
* **Bhimani, 2011 [31]** India, medical college	Pre-post trial Newly admitted medical students	17–22 y (range), 54%, 59	8 wks of 1 h/day, 5 days/wk instructor-led group **pranayama**: Kapalabhati, External Kumbhaka (Bahya), Easy Comfortable Pranayama (Sukha Purvaka), Surya Bhedan, Ujjayi, Sitkari, Sitali
* **Sharma, 2013 [70]** India, yoga research center and school	RCT Healthy healthcare students	Ix-1: 18.4 y, 77%, 30 Ix-2: 19.2 y, 87%, 30 C: 18.9 y, 83%, 30	For 12 wks, 30 m/day, 3×/week, of guided group: Ix-1: **Fast pranayama**—repeated cycles of 1 min each kapalabhati, Bhastrika, kukkriya, then 1 min rest Ix-2: **Slow pranayama**—repeated cycles of 2 min each of ANB, 6-3-6-3 breath, extended exhales w/sounds, then 1 min rest * Ix-1 and Ix-2 both end w/10 min shavasana C: No intervention
* **Sundram, 2014 [71]** Malaysia, automotive assembly plant	Pre/post-test nonequivalent-groups design Male automotive assembly line workers	20+ y (range) Ix: 20+ y, 0%, 468 C: 20+ y, 0%, 293	Ix: Five 10 min group sessions of guided **deep breathing** over 9 mo, +2×/day at-home practice encouraged C: Pamphlet on stress and its ill-effects, w/no stress reduction technique info
* **Schmalzl, 2018 [72]** US, university town	RCT Yoga-naive healthy adults	Ix: 25 y, 54%, 22 C: 24 y, 61%, 18	8-wk, 2×/wk instructor-led group classes + remaining days 15–20 min video-guided group practice: Ix: Yoga + breathing, equal-paced inhales/exhales, movements paced to breathing, 5 breaths in each pose C: Seated **breath practice**, visualizing breath moving up and down spine, directing breath to specific body parts, short breath holds after inhale/exhale, and ANB
* **Okado, 2020 [73]** US, university setting	RCT Undergraduate students	19.81 y (2.42) Ix-1: 20.16 y (3.80), 45.5%, 20 Ix-2: 19.40 y (1.75), 72.1%, 31 Ix-3: 19.68 y (1.74), 68.2%, 30 C: 19.99 y (2.37), 79%, 64	1 in-person session followed by 2 wks of individual practice: Ix-1: Guided **diaphragmatic breathing** Ix-2: Relaxation techniques, asked to identify relaxation strategies in session Ix-3: Identical to Ix-2, +activity handout w/behavioral strategies to promote relaxation C: No intervention
* **Magnon, 2021** [74] France, setting unclear	Single-arm study Healthy older and young adults	41.2 y (23.6), 80.9%, 47 Older: 65.9 y (5.1) Young: 19.6 y (1.6)	Single 5 min session of **slow breathing** following on-screen moving drop of water; starting w/4 s inhale/exhale, gradually extending exhale to 6 s
* **Balban, 2023 [75]** US, remote setting	RCT Adults w/o severe psychiatric or medical conditions	27.97 y, 68.5% Ix: 84 C: 24	28 days of daily 5 min home practice: Ix-1: **Cyclic sighing:** inhale slowly, inhale once more, slowly exhale all breath Ix-2: **Box breathing:** individualized pace w/equal length inhale, exhale, and breath holds Ix-3: **Cyclic hyperventilation w/retention:** 3 rounds of 30 deep inhales w/passive exhale, ending w/15 s hold C: Mindfulness meditation: sit/lie down, close eyes, breathe, focus on forehead between eyes
Healthy Adults: Stress/anxiety as secondary outcome			
* **Busch, 2012 [76]** Germany, university setting	Pre-post trial of 2 Ix’s separated by 6 mo Healthy undergraduate students	25 y, 87%, 15	6 weekly sessions of 20 min instructor-guided DSB, 7 bpm: Ix-1: **DSB** alone; Ix-2: DSB w/attentive concentration respiratory feedback task
Lin, 2014 [45] Taiwan, university setting	Within-subjects, 4-arm trial Healthy college students	20.98 y (1.03), 78.7%, 47	5 min sitting then **paced breathing** using on-screen pacer for 2 min per pattern w/1 min rest between Ix-1: **6 bpm, 5-0-5-0;** Ix-2: **6 bpm, 4-0-6-0;** Ix-3: **5.5 bpm, 4.4-0-6.6-0;** Ix-4: **5.5 bpm, 5.5-0-5.5-0**
* **Hunt, 2018 [21]** US, large private university	RCT Collegiate varsity soccer, swimming, track athletes	20 y, 59%, 76	10 min metronome-guided **paced breathing at 6 bpm** followed by either: Ix: **5 min guided + 5 min unguided DB**, aiming for 4 bpm C: 10 min guided PMR Afterwards, all did 5 min paced breathing (6 bpm) followed by cognitive stressor challenge
* **Naik, 2018 [77]** India, medical school	RCT Healthy adult male volunteers	Ix: 24.45 y, 0%, 49 C: 24.12 y, 0%, 50	For 12 wks: Ix: 5×/wk of 30 min guided **modified ANB** (6-6-6-0) + instructions to practice at home other 2 days/wk C: No Ix
* **Conlon, 2022 [78]** UK, University of Bath	RCT Collegiate and National Team level athletes w/no formal shooting experience	20.17 y (2.77) Ix-1: 60.9%, 23 Ix-2: 60.9%, 23 C: 57.1%, 21	Between 2 shooting task exercises, 5 min guided training + 5 min pacer-guided practice w/nasal inhales, pursed-lip exhales of: Ix-1: 4 s inhale, 6 s exhale (6 bpm) **diaphragmatic breathing** Ix-2: **2.5 s inhale, 2.5 s exhale** (12 bpm) paced breathing C: Watched 10 min respiratory anatomy educational video
* **Schlatter, 2022 [79]** France, university medical school	RCT 5th-yr anesthesiology and critical care residents participating in simulations for medical training	29 y (1) Ix-1: 50%, 12 Ix-2: 27%, 11 C: 46%, 11	For 5 min before engaging in 15 min simulated medical scenario w/performance evaluation: Ix-1: **Relaxing breathing** w/4 s inhale, 6 s exhale, following moving cursor on computer Ix-2: Same as Ix-1, but w/HRV biofeedback and instructions to increase HRV C: Read lab test results
**C. High-Anxiety Populations**			
High-Anxiety Populations: Stress/anxiety primary outcome			
* **Clark, 1990** [80] US, inpatient alcohol rehabilitation facility	RCT Male, alcohol-dependent inpatients scoring high trait anxiety (STAI > 46.62)	39.3 y Ix: 0%, 18 C: 0%, 18	Two 10 min sessions, 2 days apart, w/audio pacing tones in 1st session and no sound in 2nd: Ix: **Guided slow breathing** (10 bpm) at audio tone pace session 1; at same yet self-guided pace session 2 C: Count audio tones session 1, relax session 2
* **Chen, 2017 [81]** Taiwan, medical center	RCT High-anxiety adults from outpatient psych unit	Ix: 24 y (6), 60%, 15 C: 25 y (7), 73%, 15	12 individual training sessions over 8 wks + 2×/day home practice of: Ix: **Guided diaphragmatic breathing** C: “Routine respiration”
* **Serafim, 2018 [82]** Brazil, hospital	Pre-post open-label uncontrolled clinical trial Outpatients w/stable BD and complaints of anxiety	42.8 y, 70%, 14	Ix: 4 weekly 1:1 **guided deep breathing** trainings of 7, 10, 15, and 20 m, respectively, followed by at-home practice for 1 wk
High-Anxiety Populations: Stress/anxiety as secondary outcome			
* **Clark, 1985 [83]** UK, setting unclear	AB case series Outpatient adults with frequent panic attacks and associated anxiety	36 y, 72%, 18	Ix: 2 sessions, w/1 wk between, of 2 min voluntary hyperventilation+ discussion of over-breathing and panic attacks, followed by **guided slow breathing** Before and after Ix, 11 were exposed to individually determined anxiety-provoking situation
**D. Clinical Populations (Acute)**			
Clinical Populations (Acute): Stress/anxiety as primary outcome			
Biggs, 2003 [40] US, private dental practice	RCT Patients seeking dental treatment	38 y, 58%, 272	Written instruction in waiting room: Ix-1: **Unguided deep diaphragmatic breathing** Ix-2: Focused attention on non-teeth body part C: No intervention
* **Hayama, 2012 [84]** Japan, hospital inpatient	RCT Women w/recent Dx of gynecological cancer undergoing 1st chemo Tx	Ix: 53.6 y (9.4), 100%, 11 C: 61.7 y (9.8), 100%, 12	Ix: Before 1st chemo session and on days 2, 4, 6 post-chemo, 10 min of 4-step individual **guided diaphragmatic breathing** of 10 deep inhales and slow exhales per step, done lying down: breaths 0–10 and 30–40 w/arms raised; 11–20 abdominal and 21–30 thoracic breathing C: Usual chemo + nursing care
* **Valenza, 2014 [85]** Spain, hospital	RCT COPD exacerbation patients	Ix: 76 y, 0%, 23 C: 74.4 y, 0%, 23	Ix: 10-day, 2×/day 30 m, guided breathing of **inspiratory muscle relaxation, pursed lip breathing, prolonged and active expiration** C: Standard care
* **Bidgoli, 2016 [86]** Iran, hospital	RCT Patients undergoing first coronary angiography and experiencing anxiety	Ix: 55.5 y, 55%, 40 C: 62.7 y, 45%, 40	Ix: Single session before surgery of 5 min guided **sukha pranayama**—slow, rhythmic breathing w/equal length inhales-exhales C: Routine care
Boaviagem, 2017 [41] Brazil, maternity hospital	RCT Nulliparous women in active labor, 37–41 wks gestation	Ix: 21.2 y, 100%, 60 C: 20.6 y, 100%, 61	Ix: Instruction to use **slow deep breathing** (5 s inhales, 5 s pursed lip exhales) during dilation, active, late labor phases, including post-exhale pause (1–2 s) for active and late active labor only C: Standard care
* **Cicek, 2017 [87]** Turkey, hospital delivery room	RCT Nulliparous women 38–42 wks pregnant w/o complications in early labor	Ix: 23.3 y, 100%, 35 C: 22.4 y, 100%, 35	Ix: 30 min **guided Lamaze breath** stages for progressive labor w/real-time guidance during labor of normal nasal, slow deep chest, rapid shallow, rapid abdominal C: Routine hospital checks only
* **Chandrababu, 2019 [88]** India, hospital	Non-randomized 2-arm trial Patients undergoing 1st-time cardiac surgery	58.7 y Ix: 25%, 24 C: 29%, 24	Ix: 15 min **guided ANB**, taught 2 days prior to surgery and performed on days 3–5 post-surgery C: Usual pre- and post-operative care
Ratcliff, 2019 [47] US, university medical center	RCT Women scheduled for stereotactic breast biopsy	55.4 y (11.27) Ix-1: 55.1 y, 100%, 30 Ix-2: 55.1 y, 100%, 30 C: 55.9 y, 100%, 16	10 min prior to and throughout biopsy, guided: Ix-1: Mindfulness meditation; Ix-2: **Diaphragmatic breathing** C: Listened to neutral-content audio
* **Grinberg, 2020 [89]** US, medical center	Non-randomized pre-post study Male veterans scheduled for prostate biopsy	66.9 y (6.46) Ix: 66.7 y, 0%, 20 C: 65.05 y, 0%, 20	Ix: 15 min **guided diaphragmatic breathing** immediately prior to biopsy C: Standard care
*** Abo El Ata AB, 2021 [90]** Egypt, hospital	Pre-post study Burn patients during dressing changes	28.8 y (8.9), 37%, 7	2 wk daily **guided relaxation breathing** exercises before and during daily dressing change
* **Hosseinzadeh-Karimkoshteh, 2021 [91]** Iran, medical center	RCT Burn patients	Ix: 25.5 y, 33.3%, 15 C: 28.1 y, 40%, 15	4 days of: Ix: 30 min **guided relaxation breathing** before dressing change, 4-4-4-0 pursed-lip exhales C: Routine care
* **Zahn, 2021 [92]** Switzerland, university hospital	RCT Adults having dermatological surgery w/local anesthesia	Ix: 60.7 y (20.6), 40.8%, 86 C: 60.2 y (18.9), 48.8%, 84	Before undergoing surgery: Ix: **Deep breathing instruction**, were told to perform before and during procedure C: No Instruction
* **Moghadam, 2022 [93]** Iran, psychiatric hospital	RCT Adults receiving ECT for depression	Ix-1: 35 y (14), 50%, 30 Ix-2: 38 y (9), 53.3%, 30 C: 36 y (10), 50%, 32	Before ECT procedure: Ix-1: 10 min **guided slow breathing**, (5 s deep nasal inhale, 5 s mouth exhale) Ix-2: 3–5 min breathing lavender aromatherapy oil C: Routine care
* **Aktas, 2023 [39]** Turkey, Ankara Hospital	RCT Postoperative bariatric surgery patients	34.7 y Ix-1: 76.7%, 23 Ix-2: 66.7%, 20 C: 53.3%, 16	Each h during post-op h’s 1-6: Ix-1: **4-7-8 breathing technique**, 10 breaths; Ix-2: **Deep Breathing,** 4 breaths C: Treatment as usual
Clinical Populations (Acute): Stress/anxiety as secondary outcome			
* **Dhruva, 2012 [94]** US, university medical center	RCT w/single-arm crossover Cancer patients undergoing intravenous chemo	Ix: 52 y, 75%, 8 C: 56 y, 100%, 8	During 2 consecutive chemo cycles: Ix: Usual care + 60 min class 1×/wk + 2×/day of 10–15 min **individual home pranayama**: * breath observation * ujjayi breathing: slow, deep abdominal rhythmic inhales/exhales, w/extended exhales and partially closed glottis * kapalabhati breathing: gentle inhales, brief pause, forceful exhale * ANB C: Usual care for 1st chemo cycle; usual care + Ix for 2nd cycle
* **Park, 2013 [95]** South Korea, medical center	Pre-post study comparison group Burn patients	44.9 y Ix: 44.5 y, 43.3%, 30 C: 45.3 y, 60%, 30	Over 3 days: Ix: **Guided relaxation breathing** 4-0-4-0 before and 4-0-2-0 during dressing changes C: Usual dressing change procedure
* **Eldin, 2015 [96]** Libya, hospital	RCT Burn patients	Ix: 18–50 y, 45%, 20 C: 18–50 y, 30%, 20	Over 3 days: Ix: 15+ min **guided relaxation breathing** (in-nose, out-mouth) C: Standard care
* **Iyer, 2020 [97]** India, medical center	Pre-post Burn patients	31.8 y (17.8), 35%, 20	Over 7 days: Ix: 15 m/day **guided deep breathing**
* **Ursavas, 2020** [98] Turkey, hospital	RCT Total knee replacement patients	Ix: 65 y, 79%, 19 C: 69 y, 79%, 19	Ix: **Guided diaphragmatic breathing** + instruction to use at 1, 2, 4, 8, 12, and 24h post-op + when experiencing pain/anxiety C: Standard care
* **Lu, 2022 [99]** China, hospital	RCT Lung cancer patients undergoing thoracic surgery	59.3 y (10.2) Ix: 55%, 34 C: 61%, 34	Over 14 days: Ix: 2×/day of 15–20 min **guided Active Cycle of Breathing Technique**, each part repeated 3–5×: slow inhale, 3 s hold, 6 s pursed-lip exhale; deep active inhale, 3 s hold, passive exhale (all nasal; slow deep nasal inhale, 2–3 forceful “huff” mouth exhales) C: Usual care
**E. Clinical Populations (Chronic)**			
Clinical Populations (Chronic): Stress/anxiety as primary outcome			
* **Marshall, 2014 [100]** US, setting unclear	Pre-post repeated-measures trial Post-stroke individuals w/brain damage, w and w/o aphasia	55.6 y, 18%, 11	4 wks of 1 h guided practice 1×/wk, followed by 6 wks of 5–40 min independent practice of **right UNB**: occlude left nostril while inhaling and exhaling through the other, work toward exhaling twice as long as inhale
Clinical Populations (Chronic): Stress/anxiety as secondary outcome			
* **Han, 1996 [22]** Belgium, setting unclear	Pre-post single-arm trial Adults w/HVS	37 y, 65%, 92	17 sessions over 2.5 mo of physiotherapist-guided breathing therapy and retraining: * 1st session: 3 min **voluntary hyperventilation** at 60 bpm followed by therapy to reattribute cause of daily life complaints to hyperventilation and explain breath retraining purpose * Remaining sessions: 45 min breath retraining w/**abdominal breathing and slowed exhales**
* **Thomas, 2009 [101]** UK, primary care general practices	RCT Adults w/asthma and impaired health status	Ix: 46 y (median), 58%, 66 C: 46 y (median), 64%, 63	Ix: 3 physiotherapist-guided sessions w/2–4 wks between each: * One 1 h small group breathing education session * Two 30–45 min individual **DB and nasal breathing** training sessions * Participants instructed to practice breathing exercises ≥10 m/day C: Asthma nurse-delivered education
Jefferson, 2010 [43] US, setting unclear	RCT African-American females w/hypertension	Ix: 54 y, 100% C: 56 y, 100%	Ix: 20 min therapeutic chair massage C: Education on **DB**
* **Sureka, 2014 [102]** India, prison hospital	RCT Adult male prisoners w/psychiatric disorder	Ix: 36 y, 0%, 116 C: 36 y, 0%, 115	For 6 wks, 30 m/day: Ix: Guided **SKY breathing**: 8 min Ujjayi, 5 min bellows (Bhastrika) followed by Om Chant, 10 min SK, and 5 min ANB C: Seated w/eyes closed and gentle attention on breath
* **Sureka, 2015 [103]** India, prison hospital	RCT Male inmates w/substance dependence	Ix: 39.3 y, 0%, 55 C: 38.8 y, 0%, 56	For 6 wks, 23 m/day: Ix: Guided **SKY breathing:** 8 min Ujjayi, 5 min bellows (Bhastrika), and 10 min SK C: Sitting, eyes closed, attention to breath
Thomas, 2017 [50] UK, general practices	RCT Adults w/impaired asthma control	Ix-1: 56 y, 63%, 261 Ix-2: 55 y, 69%, 132 C: 57 y, 63%, 262	Ix: Guided **breath training** delivered either via DVD (Ix-1) or 3 biweekly 30–40 min sessions w/physiotherapist (Ix-2), comprising: nose breathing, abdominal breathing, slow breathing, controlled breath holding, relaxation C: Usual care
* **Fiskin, 2018 [104]** Turkey, medical center	RCT Pregnant women w/GDM	Ix: 30.6 y, 100%, 30 C:31.3 y, 100%, 30	30 days of: Ix: Unguided **diaphragmatic breathing**, 5 min in the morning before leaving bed C: Standard care, invited to talk about their pregnancies 2×/mo
* Watson, 2022 [105] New Zealand hospital	RCT (cross-over) Postmenopausal women w/stress cardiomyopathy	66.8 y (2.7) Ix: 100%, 12 C: 100%, 12	One 3 min session each, separated by 5 days, of guided: Ix-1: **Hyperventilation;** Ix-2: **Slow breathing w/3 s hold**
**F. Simulated Stress**			
Simulated Stress Populations: Stress/anxiety as primary outcome			
Kamath, 2017 [44] India, medical school	RCT Healthy, yoga-naive undergrad medical students	Range 19–24 y Ix: 60%, 15 C: 80%, 15	After brief instruction and practice for 15 min prior to **simulated public speaking** task: Ix: Guided **ANB** C: Sitting in quiet room
Simulated Stress Populations: Stress/anxiety as secondary outcome			
Holmes, 1978 [42] US, university setting	3-arm clinical trial (randomization/Tx allocation unclear) Undergraduate students	Age not reported, 54%, 111	90 s threat of electric shock (never administered) or visual stimulation, after 5 min of: Ix-1: **Respiration tracing**, breathing at pace where real-time respiration polygraph pen matched previously recorded resting respiration polygraph Ix-2: Attention tracing, using hand-held knob to follow previously recorded resting respiration polygraph recording C: Sitting quietly and relaxing
* **McCaul, 1979 [106]** US, university setting	RCT, 3 × 2 + 1 factorial-designed Male undergraduate students	Ix-1: 0%, 29 Ix-2: 0%, 28 Ix-3: 0%, 28 C: 0%, 14 (Ages not Reported)	3 groups threatened w/impending electric shock, w/possibly 2 min (duration unclear) of unguided: Ix-1: **Slow-pace regulated breathing** (8 bpm); Ix-2: Normal-pace regulated breathing (16 bpm); Ix-3: Non-regulated breathing C: No breath pacing, no expectations, no threat, anticipation of visual stimulation Ix-1 and Ix-2 had 2 min practice before threat
Sakakibara, 1996 [48] Japan, university setting	RCT Healthy college students	23 y Ix-1: 40%, 10 Ix-2: 30%, 10 C: 50%, 10	Prior to 2 min threat (electric shock) anticipation w/15 bpm paced breathing, 5 min of unguided: Ix-1: **Slow breathing (8 bpm)** Ix-2: **Fast breathing (30 bpm)** C: Spontaneous breathing
* Telles, 2019 [49] India, residential yoga center	Randomized cross-over study Adult males w/yoga breathing experience	28.4 y (8.2), 0%, 50	In 3 separate, 18 min sessions across 3 days, w/timed shape and size discrimination task before/after each: Ix-1: **ANB** Ix-2: Breath awareness C: Quiet sitting
Meier, 2020 [46] Canada, setting unclear	RCT Young healthy men and women	23.8 y (4.5), 51%, 35 Ix-1: 12 Ix-2: 11 C: 12	30 m, prior to simulated stress test (modified TSST-G): Ix-1: Guided **relaxation breathing** while standing Ix-2: Laughter yoga C: Reading magazines or books
*** Sharma, 2022 [107]** India, university setting	Randomized cross-over study University males w/yoga breathing experience	24 y (4), 0%, 38	6 breath sessions in random order, w/timed SLCT before/after each, for 18 min w/1 min rest every 5 m, guided: Ix-1: **ANB;** Ix-2: **Bellows breathing;** Ix-3: **Bumblebee breathing;** Ix-4: **High-frequency breathing at 50–65 bpm;** Ix-5: Breath awareness; Ix-6: Quiet seated rest

Interventions in red font were ineffective for stress outcome (i.e., did not significantly reduce participants’ stress/anxiety). Bolded phrases in “Intervention and Control” column reflect key practice details. Bolded author names/years with asterisk (*) represent studies with at least one intervention that did significantly reduce participants’ stress/anxiety. ANB, alternate-nostril breathing; BD, bipolar disorder; bpm, breaths per minute; chemo, chemotherapy; C, control; chemo, chemotherapy; COPD, chronic obstructive pulmonary disease; dept, department; DSB, deep slow breathing; DB, diaphragmatic breathing; DVD, digital versatile disc; Dx, diagnosis; ECT, electroconvulsive therapy; GDM, gestational diabetes mellitus; h, hour; HR, heart rate; HVS, hyperventilation syndrome; Ix, intervention; min, minutes; mo, month; N, number of subjects; PMR, progressive muscle relaxation; RCT, randomized controlled trial; s, seconds; SD, standard deviation; SK, Sudarshan Kriya; SLCT, Six-Letter Cancellation Test; TSST-G, Trier Social Stress Test for Groups; Tx, treatment; UNB, unilateral nostril breathing; vs., versus; w/, with; w/o, without; wks, weeks; y, years. Practices with numerical annotations such as 4-4-4-0 reflect [inhale count]-[hold count]-[exhale count]-[hold count].

**Table 2 brainsci-13-01612-t002:** Association of intervention parameters with intervention stress/anxiety reduction effectiveness among 72 included interventions.

Parameter	N	*p*-Value
Study design		
Study design (RCT vs. other)	72	0.77
Control group (yes/no)	72	0.19
Primary vs. Secondary stress/anxiety outcome metric	72	0.78
# stress/anxiety outcome measurements (1, 2, 3+)	72	0.43
Breath practice and implementation		
Population	72	0.14
Breath practice pace	72	0.01 *
Breath pace other than fast-only	70	0.10
Human-guided training (yes/no)	72	<0.001 *
# human-guided sessions (0, 1, 2−3, 4+)	71	<0.001 *
Human-guided training included within device-guided interventions (yes/no)	15	0.03 *
Single vs. Multiple sessions	72	<0.001 *
Long-term vs. No long-term practice	72	0.005 *
Intervention duration (1, 2−5, 6−30, or 31+ days)	71	<0.001 *
Session duration:		
<5, 5–10, 11–20, or >20 m	61	0.03 *
<5 or ≥5 m	61	0.005 *
Among sessions ≥5 m: 5–10, 11–20, or >20 m	52	0.81
Group vs. individual practice vs. both	67	0.79
At-home practice (yes/no)	72	0.24

N reports the number of interventions in which parameter value was reported and thus the number of interventions included in the corresponding Fisher’s exact test. *p*-values based on Fisher’s exact tests on two-way frequency tables of effective versus ineffective interventions that either did or did not implement each parameter (Appendix A shows frequency counts; Appendix A shows statistical tests). *p*-value ≤ 0.05 signifies that intervention effectiveness was not associated with that parameter. Highlighted parameters identified as components for conceptual framework of factors associated with effectiveness in the 72 stress reduction breathing interventions among the 58 studies included in this review. Control categories defined as follows: yes, active and/or non-active control; or no, no control or another breath intervention only. Population categories defined as youth, healthy adults, high-anxiety populations, clinical populations (acute), clinical populations (chronic), and simulated stress populations. Breath practice pace defined as slow only, fast only, combined fast and slow with or without ANB/UNB, and ANB/UNB alone. Human-guided training defined as use of live or pre-recorded audio or video human instruction throughout at least the initial breath session. Multiple sessions defined as performing breathing practice more than once. Long-term practice defined as performing ≥6 sessions over ≥1 week. m, minutes; vs., versus; #, number. * Statistical significance at *p* < 0.05.

**Table 3 brainsci-13-01612-t003:** Outlier interventions with explanatory practice and research caveats.

Intervention		Potential Caveats	
Author, Year (Population)	Description	Practice	Research
Effective practices incorporating zero core components			
Schlatter, 2022 [79] (healthy adults)	5 min of 6 bpm relaxing breathing prior to simulated medical scenario and test	Investigator present during device-guided practice Physiologically trained residents	
Ineffective practices that incorporated the following components:			
Human-guided training			
Meier, 2020 [46] (simulated stress)	30 min guided standing relaxation breathing prior to simulated stress test	Standing still Interrupted practice	
Human-guided training, multiple sessions			
Kamath, 2017 [44] (simulated stress)	15 min guided ANB prior to simulated public speaking task	Highly technical practice ANB task performance effects may attenuate stress-reducing effects	
Ratcliff, 2019 [47] (clinical, acute)	Guided diaphragmatic breathing for 10 min prior to and during breast biopsy	Difficult to execute lying on stomach and remaining still for biopsy	Practice guides not blinded
Human-guided training, multiple sessions, long-term practice			
Thomas, 2017 [50] (clinical, chronic)	Guided diaphragmatic breathing with breath holds, guided via DVD		Practices not continued through outcome period Outcomes reported at baseline and 12 mo
	Same as above, delivered by physiotherapist		

Outliers defined as effective interventions that incorporated zero core components and ineffective interventions that incorporated ≥1 core component and satisfied both exclusionary components (e.g., avoided both fast-only breathing and sessions <5 min). Caveats report breath practice and study-specific research characteristics that may have increased interventions’ stress-reducing effects in effective outlier interventions and those that may have lessened or diluted these effects in ineffective outlier interventions. ANB, alternate-nostril breathing; bpm, breaths per minute; DVD, digital versatile disc; min, minutes; mo, months; w/o, without.

**Table 4 brainsci-13-01612-t004:** Actionable takeaways.

For Practitioners, Program Developers, and Leaders Designing and Implementing Breath Practices	For Researchers Designing and Implementing Breath Practice Studies
Pace: ◦ Choose any breath pace that incorporates some form of regulated slow breathing, such as: ◾ extended exhales; box breathing; slow breathing with pauses/holds; slow diaphragmatic breathing; combined fast+slower breathing; ANB/UNB ◦ Avoid practices involving fast breathing alone Training: ◦ For at least the 1st session, follow human-voice guidance ◦ This can be in-person or via live or recorded video/audio ◦ It can be as short as 5 min ◦ Get extra training if doing ANB/UNB for the first time Setup: ◦ Be in a comfortable seated or lying down position ◦ Avoid extensive standing ◦ Have access to free diaphragmatic movement ◦ Be in a place where you will not be interrupted Practice: ◦ Avoid practices shorter than 5 min ◦ Do it more than once ◦ Keep doing it for continued benefit	Breath practice design: Follow the practice tips in left-hand column Study design: ◦ For real-time practice guidance, use instructors or guides who are blinded to experimental condition ◦ Create accountability for, motivate, and assess practice compliance, especially for: ◾ home practices ◾ longer-term follow-up periods ◦ Ensure that practices are continued throughout outcome measurement periods

Extended exhale breathing defined as a breath pattern in which the exhale is longer than the inhale; box breathing defined as breathing with equal durations of inhales, post-inhale holds, exhales, and post-exhale holds. ANB, alternate-nostril breathing; UNB, unilateral nostril breathing.

## Data Availability

Data synthesis and output available in Appendix A. Other data available on request.

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
