# Peer review of "Breathing Practices for Stress and Anxiety Reduction: Conceptual Framework of Implementation Guidelines Based on a Systematic Review of the Published Literature"

_brainsci, 2023, doi:10.3390/brainsci13121612_

Round 1
Reviewer 1 Report
Comments and Suggestions for Authors
Thank you for the opportunity to review this manuscript. I rarely review papers where I don't have much to critic. I thought the authors did a good job detailing their methodological approach and systematically presenting the findings. I also agree with their assessment of the quality of studies and how it impacts their ability to conduct this analysis (in the limitations and future directions section). This shortcoming becomes apparent when conducting a systematic review of findings and meta-analyses, as it always presents a unique challenge to the researchers. I thought the authors did a good job navigating this issue. I do have a couple of observations and recommendations for tables below.
Table 1 is obviously vitally important. However, the formatting makes some of it hard to read as some of the sections are long and word wrap in a manner that breaks up words. For example, the stress metric column under table 1B. Bhimani 2011 is only allotted up to 6 characters width, and so the word questionnaire is broken into 3 separate rows to fit.
Figure 5 is interesting, but it is harder to read due to the small size. Any way to break it up or reduce some of the graphics so the displayed information can be a little larger?
Figure 6 does a good job summarizing findings, I found this useful.
Reviewer 2 Report
Comments and Suggestions for Authors
The article deals with a very interesting topic related to breathing practices for stress reduction. The systematic review study is well planned and conducted. However, some points need to be corrected. In general, the study is very interesting, but some sections should be improved.
1. Please include the background in the abstract section.
2. How many documents, authors, journal sources, references, and keywords were retrieved and included in this study? Please include it in the “abstract” section.
3. Please enter all inclusion criteria consecutively in the “method” section.
4. Please place the table title at the top of the table.
5. For better understanding, please explain the meaning of “blue and red *” below Figure 5.
6. Please indicate the meaning of “*” below Tables 2 and 3.
7. Please include the appropriate reference in the section.
“Common voluntary regulated breathing practices include diaphragmatic breathing, paced slow breathing, breathing with biofeedback, and alternate nostril breathing (ANB). Breathing practices, when used in isolation, have the advantage of being universally accessible, scalable, and cost-free. They are not limited by access to healthcare services nor burdened by side effects, and put potential treatment tools in the hands of the individual.”
8. Please improve the quality of figure 3.
9. Please include the “Acknowledgment” section.
Comments on the Quality of English LanguageMinor editing of English language required
Reviewer 3 Report
Comments and Suggestions for Authors
this paper is very difficult to read because of its organization. the paper is hybrid between narrative review and systematic review. some terms used are inadequate e.g. automatic searches = electronic search and poor choice of databases e.g. PubMed and ScienceDirect show that conduct of systematic review was poorly handled. Table 1 and Table 2 tells readers very little only p value with no effect size and direction makes this data uninterpretable.
Prisma flowchart show artificial numbers e.g. articles unobtaiable12 contradict that they were full text screened.
Previous/recent meta analysis showed low effect size https://www.nature.com/articles/s41598-022-27247-y while authors cited this work they did not interpret it correctly.
Reviewer 4 Report
Comments and Suggestions for Authors
Thanks to the authors for sharing their research. I think this is an interesting study, but I have some concerns:
1. In the search strategy, the authors note that they used PubMed and ScienceDirect to search for relevant literature. It is unclear to me what justifies such a choice, why other important databases (EMBASE, Cochrane, Web of Science and PsycINFO) were not used. For a more complete systematic review, I recommend that the authors expand the databases.
2. The title of the article reads as follows: «Breathing practices for stress reduction: conceptual framework of implementation guidelines based on a systematic review of the published literature». Meanwhile, in the outcomes, the authors studied not only stress, but also anxiety. Further in Table 1, they call the column «stress metrics», but describe instruments measuring anxiety (for example, HADS-anxiety). It seems to me important to pay more attention to the terms and not exclude anxiety from the names, since it was used in a systematic review.
3. The authors analyzed studies involving people from 1 year old. It seems to me that such a restriction is unacceptable, even though the authors further describe youth and healthy adults separately. Why were there no maximum age restrictions? Maybe if there were no such restrictions, then it is also necessary to separately represent a group of elderly people in the sample of healthy adults?
These questions and comments seem to me significant and require a major revision of the manuscript.
Round 2
Reviewer 3 Report
Comments and Suggestions for Authors
My concerns remain unresolved. My two concerns were not possible to be addressed by simple rewrite and major changes were needed in order to reevaluate the paper.
The largest psychiatry database is omitted = APA PsycInfo and the most comprehensive database is not included also = Scopus.
The rebuttal “ To account for lack of additional databases, 20 key studies, most of which were reviews that together searched a range of other databases, were manually reference-mined for additional articles”. Is a departure from a systematic review process, as it does not account for your specific PICOs. Relying on another reviews/studies is a secondary approach not primary.
The second critical issue is that “Table 1 and Table 2 tells readers very little, with only p value with no effect size and direction makes this data uninterpretable”.
The rebuttal provided is not justified effect sizes need be visible if not meta-analyses “The effect sizes would not be truly comparable due to heterogeneity in the included studies of tests used for measuring stress/anxiety. For example, some studies used visual scales while others used more sophisticated questionnaires, and the scale ranges and granularity varied largely. Relative effect sizes would be largely uninterpretable”.
Furthermore, heterogeneity is assumed here and not documented as per standard approaches i2 or tau2, and not sure if you mentioning within or between studies heterogeneity. The argument provided by the authors as a rebuttal suggesting that this can be dealt as subgroups with visual scales vs. more sophisticated questionnaires which I believe you refer to “psychometrically” sound measures is needed.
For quite a few studies, effect sizes were not computable from the information provided (e.g., when studies reported only mean differences, without
The authors stated “Some studies compared more than 2 groups or interventions, and/or used more than one tool to measure the same outcome. These studies would contribute more than one effect size, rendering those effect sizes not truly independent”. I don’t understand how authors justify presenting p value and avoid effect size on this basis.
I tried to read Table 2 again and using “Human-guided training (yes/no)” with studies = 72 and p value 0.001 the authors explained that most studies suggesting that this parameter is associated with improvement. Exception was Meier 2020. I remained with many questions here why? How about measurement error? 95%CI why can’t readers see if Miller 95%ci is actually passing positive size?
My aim is to make the paper stronger and present information that reviewers and readers can digest. In the present form I think only authors have the full information/understanding and we are we remain unable to unbox it.
Author Response
My concerns remain unresolved. My two concerns were not possible to be addressed by simple rewrite and major changes were needed in order to reevaluate the paper.
We appreciate this reviewer’s time in reviewing our revised manuscript.
1. The largest psychiatry database is omitted = APA PsycInfo and the most comprehensive database is not included also = Scopus. The rebuttal “ To account for lack of additional databases, 20 key studies, most of which were reviews that together searched a range of other databases, were manually reference-mined for additional articles”. Is a departure from a systematic review process, as it does not account for your specific PICOs. Relying on another reviews/studies is a secondary approach not primary.
As explained in detail in our prior response, one very practical reason for this choice was our very limited resources for this independent, unfunded, and 100% volunteer-run study. At this time, we do not have the human, time, or financial resources to redo the search, screen, and abstraction process with 2 more databases; in addition, as independent researchers we do not have access to the particular databases requested. The insistence that we use databases accessible by larger and/or better funded institutions creates a bias against publication of research done by smaller entities. Many systematic reviews search only a few databases; despite this constraint in our study, we have included a high number of articles that is magnitudes greater than that of other reviews on the topic, lending significant credence to our conclusions. We have added to the manuscript the use of 2 databases as a limitation, as follows (page 35, lines 843-845):
“Only 2 databases were searched for this systematic review; however, the large number of identified and included studies lends credence to the conclusions presented.”
2. The second critical issue is that “Table 1 and Table 2 tells readers very little, with only p value with no effect size and direction makes this data uninterpretable”. The rebuttal provided is not justified effect sizes need be visible if not meta-analyses “The effect sizes would not be truly comparable due to heterogeneity in the included studies of tests used for measuring stress/anxiety. For example, some studies used visual scales while others used more sophisticated questionnaires, and the scale ranges and granularity varied largely. Relative effect sizes would be largely uninterpretable”. Furthermore, heterogeneity is assumed here and not documented as per standard approaches i2 or tau2, and not sure if you mentioning within or between studies heterogeneity. The argument provided by the authors as a rebuttal suggesting that this can be dealt as subgroups with visual scales vs. more sophisticated questionnaires which I believe you refer to “psychometrically” sound measures is needed. For quite a few studies, effect sizes were not computable from the information provided (e.g., when studies reported only mean differences, without
While effect size would be relevant if our purpose were to evaluate overall effectiveness of included interventions, we instead aimed to evaluate the relationships between intervention parameters and effectiveness, with the goal of building the framework described in the paper. Accordingly, our results are not based on effect sizes of interventions: rather, they are based on the odds that X number of interventions would have an effect while implementing a specific tested parameter, or in other words, the odds ratio being different than one. To test this we have used Fisher’s exact test, and we report the resulting p-values of these tests in Table 2, as well as more detailed odds ratios and 2-way frequency tables in supplemental material. Therefore, the FET p-values are presented without the effect sizes of each individual intervention. In fact, showing effect sizes would detract more than it would enhance our paper, creating clutter and confusion in tables that are already long and contain detailed intervention information relevant to our analysis.
The authors stated “Some studies compared more than 2 groups or interventions, and/or used more than one tool to measure the same outcome. These studies would contribute more than one effect size, rendering those effect sizes not truly independent”. I don’t understand how authors justify presenting p value and avoid effect size on this basis.
It is our understanding that the request is to show pooled effect sizes, since they are requested to be seen in Table 2. Thus the above response is relevant, because pooling the individual effect sizes which are not truly independent could bias the pooled effect size.
I tried to read Table 2 again and using “Human-guided training (yes/no)” with studies = 72 and p value 0.001 the authors explained that most studies suggesting that this parameter is associated with improvement. Exception was Meier 2020. I remained with many questions here why? How about measurement error? 95%CI why can’t readers see if Miller 95%ci is actually passing positive size?
We were also left wondering why – which is why we sought to find explanatory caveats in this study and have reported them in the outliers table.
My aim is to make the paper stronger and present information that reviewers and readers can digest. In the present form I think only authors have the full information/understanding and we remain unable to unbox it.
Reviewer 4 Report
Comments and Suggestions for Authors
Dear authors, thank you for your attention to the comments!
Author Response
Dear Reviewer,
Thank you for your response. We are pleased that you were happy with our revisions.
We appreciate your time and energy in reviewing this work.